# Facing obesity in pain rehabilitation clinics: Profiles of physical activity in patients with chronic pain and obesity—A study from the Swedish Quality Registry for Pain Rehabilitation (SQRP)

**Huan-Ji Dong**[1]*, **Britt Larsson**[1], **Marcelo Rivano Fischer**[2,3], **Björn Gerdle**[1]

1 Pain and Rehabilitation Centre, and Department of Health, Medicine and Caring Sciences, Linköping University, Linköping, Sweden, 2 Department of Health Sciences, Research Group Rehabilitation Medicine, Lund University, Lund, Sweden, 3 Department of Neurosurgery and Pain Rehabilitation, Skane University Hospital, Lund, Sweden

* huanji.dong@liu.se

**Data Availability Statement:** The datasets generated and/or analysed in this study are not

## Abstract

### Background

The obesity epidemic has influenced pain rehabilitation clinics. To date, little is known about baseline level of physical activity (PA) in patients referred to pain rehabilitation clinics. We aimed to investigate the PA levels of patients referred to pain rehabilitation clinics and to evaluate the effect of excess weight on PA level.

### Methods and findings

Data were obtained from the Swedish Quality Registry for Pain Rehabilitation between 2016 and 2017. These data included PA time (everyday PA and physical exercise per week), Body Mass Index (BMI), sociodemographic factors, chronic pain and psychological aspects (e.g., pain intensity, depressive and anxiety symptoms and insomnia problems). Insufficient PA was defined as less than 150 minutes per week. We performed logistic regressions as well as orthogonal partial least square regression to estimate the effects of excess weight on PA. Over one-fourth of the patients were classified as obese (BMI $\geq$30 kg/m$^2$, 871/3110, 25.3%) and nearly one-third of these patients were classified as severely obese (BMI $\geq$35 kg/m$^2$, 242/871, 27.8%). Time estimations for physical exercise varied among the BMI groups, but patients in the higher BMI category were more likely to spend less time on every-day PA. Compared to normal weight, mild obesity [odds ratio (OR) 0.65, 95% confidence interval (CI) 0.53–0.81] and severe obesity (OR 0.56, 95% CI 0.42–0.74) were associated with less PA. Mild obese patients had an elevated risk of 65% and severe obese patients had an elevated risk of 96% for insufficient PA. Increased pain intensity was positively related to insufficient PA (OR 1.17, 95% CI 1.06–1.29) among the obese patients.

publicly available as the Ethical Review Board has not approved the public availability of raw data. The data that support the findings of this study are available from SQRP (https://www.ucr.uu.se/nrs/) but restrictions apply to the availability of these data, which were used under license for the current study, and so are not publicly available. Data are however available upon reasonable request and with permission of SQRP research group (address: NRS, Skånes Universitetssjukhus, Smärtrehabilitering, Lasarettsgatan 13, SE 221 85 Lund, Sweden; Register holder: Marcelo Rivano Fischer, Marcelo.rivanofischer@skane.se). Analyzing data also requires permission from the Swedish Ethical Review authority (In Swedish: Etikprövningsmyndigheten; address: Etikprövningsmyndigheten, Box 2110, SE 750 02 Uppsala, Sweden, e-mail: registrator@etikprovning.se).

**Funding:** This study was supported by grants from AFA insurance, the County Council of Östergötland (Research-ALF, LIO-608021, BG and SC-2017-00202-28, H-JD). AFA Insurance, a commercial founder, is owned by Sweden's labor market parties: The Confederation of Swedish Enterprise, the Swedish Trade Union Confederation (LO), and The Council for Negotiation and Co-operation (PTK). These parties insure employees in the private sector, municipalities and county councils. The funding sources did not participate in the design of the study, collection, analysis, or interpretation of the data; or in the decision to submit the manuscript for publication. There was no additional external funding received for this study.

**Competing interests:** AFA Insurance provided support in the form of a research grant, which was used for salaries for the author BG and for language revision, publication costs and participation in scientific conferences. This does not alter our adherence to PLOS ONE policies on sharing data and materials.

## Conclusion

Having low PA is very common for patients referred to pain rehabilitation clinics, especially for those with comorbid obesity. As a first step to increase PA, obese patients need to be encouraged to increase the intensity and amount of less painful daily PA.

## Introduction

Both chronic pain and obesity are significant health concerns. In Europe, about 20% of the adult population report suffering moderate to severe chronic pain [1]. Obesity, a common comorbidity of chronic pain, has been described as a global pandemic [2, 3]. A survey of over a million individuals in the United States reported pain incidence to be 68% to 254% more frequent in people classified as obese (mild to severe obesity) than in people with low to normal weight [4].

In everyday practice in pain rehabilitation clinics, patients with excessive weight burden are prevalent. Although the relationships between pain and obesity have not been fully explored, a growing body of literature strongly suggests that pain and obesity negatively impact each other [4–8]. Patients report a series of mutual health consequences such as physical limitation [9, 10], low psychological wellbeing [11], sleep disturbances [12, 13], poor health-related quality of life (HRQoL) [14, 15], and function dependence [16, 17].

As expected, patients suffering chronic pain and obesity engage in less physical activity (PA) as both conditions appear to be associated with physical deconditioning [8, 18]. Higher levels of PA have been related to lower pain, less fatigue, and better quality of life [16, 19, 20]. Moreover, PA is recognized as a health-promoting behaviour for the general population as well as for patients with chronic conditions [21, 22]. Increasing evidence shows that prescribed PA, a non-pharmacological intervention, contributes to reducing chronic pain and body fat [23, 24]. These research findings address not only the importance of focusing on patients' non-optimal lifestyle in pain management, but also the promising benefits of improving health behaviours.

In Sweden, interdisciplinary multimodal pain rehabilitation programs (IMMRPs) for chronic pain, which are well-coordinated interventions according to the International Association for the Study of Pain (IASP), include activities such as pain education, supervised PA, training in simulated environments, and cognitive behavioural therapy [25]. Typically, health care providers in Sweden encourage all their patients to decrease their sedentary time and to increase their PA. However, patients, especially overweight/obese patients with chronic pain [26], often complain that their pain has been the major barrier to their PA. Hence, there appears to be a vicious circle: pain results in less PA, less physical PA in more body weight, and more body weight results in more pain [7, 11]. In clinical settings, patients assessed at specialist pain rehabilitation clinics report complex pain-related conditions and comorbidities that might negatively affect their PA. Surprisingly, despite the high prevalence of overweight and obese patients in pain rehabilitation clinics, baseline data rarely includes the PA of the patients [25, 27, 28].

To address the above gap in knowledge, this study uses the Swedish Quality Registry for Pain Rehabilitation (SQRP) database to investigate the levels of self-reported PA of chronic pain patients referred to specialist pain rehabilitation clinics. In addition, this study investigates the impact of reported excess weight on PA (effect modification determined by obesity stratification).

## Materials and methods

### The Swedish Quality Registry for Pain Rehabilitation (SQRP)

The SQRP is recognized by the Swedish Association of Local Authorities and Regions. Approximately 40 public and private clinics, equating to > 90% coverage of the clinical departments offering pain rehabilitation at the specialist level in Sweden, send data to the SQRP [28]. The SQRP uses questionnaires to capture patients' sociodemographic background, pain characteristics, psychological symptoms, function, activity/participation aspects, and Health-related Quality of Life (HRQoL). A more detailed description of the registry has been reported in previous publications [27]. In 2010, the Boston Consulting Group ranked the SQRP as one of the ten high-quality national registries in Sweden [29].

### Subjects

This cross-sectional study includes patients referred to pain rehabilitation clinics between August 2016 and February 2017 for clinical assessments, medical treatment, and rehabilitation. These patients were ≥ 18 years old and had non-malignant chronic pain (≥ 3 months) usually reported as part of complex chronic pain conditions that required a bio-psycho-social assessment as well as intervention.

All patients in SQRP gave their written informed consent and signed a consent form that was in accordance with the Declaration of Helsinki. This study was approved by the Linkoping University Ethics Committee (Dnr: 2015/108-31).

### Sociodemographic aspects and body weight

The following sociodemographic characteristics were selected from the SQRP: age (years); sex (men or women); highest education levels (university/college, upper secondary school, and elementary school dichotomized into university/college and the other alternatives); place of birth (country outside Europe, Europe non-Nordic country, Nordic country and Sweden dichotomized into Europe vs. outside Europe); and work/study status (yes or no).

We calculated BMI ($kg/m^2$) using self-reported weight and height in the SQRP and classified BMI according to the World Health Organization (WHO) criteria: <18.5 = underweight; 18.5–24.9 = normal range; 25.0–29.9 = overweight; 30.0–34.9 = obesity class I, mild obesity; and ≥35.0 = obesity class II-III, severe obesity.

### Pain aspects

Pain duration was determined using the patients' report of when they first experienced their current pain (days). The number of months were then calculated by dividing by 30. Pain intensity during latest seven days (NRS-7d) was determined using the patients' report of their pain intensity during the previous week using a numeric rating scale (NRS) with 0 representing no pain and 10 representing worst possible pain. Pain frequency was determined using the patients' report of their pain as either constant or recurrent. This variable was denoted as constant pain or no constant pain. Pain distribution was determined using the Pain Region Index (PRI). The PRI consists of 36 predefined anatomical areas (18 on the front and 18 on the back of the body). The number of areas with pain were calculated by adding all the patients' marked anatomical areas that represent where they experience pain: 1) head/face, 2) neck, 3) shoulder, 4) upper arm, 5) elbow, 6) forearm, 7) hand, 8) anterior aspect of chest, 9) lateral aspect of chest, 10) belly, 11) sexual organs, 12) upper back, 13) low back, 14) hip/gluteal area, 15) thigh, 16) knee, 17) shank, and 18) foot.

## Psychological aspects

The Hospital Anxiety and Depression Scale (HADS) is a self-assessment questionnaire that measures anxiety and depression symptoms [30]. HADS is divided into an anxiety subscale (HAD-A) and a depression subscale (HAD-D). Both subscales have seven items with a scoring range of 0 to 21; the lower score indicates a lower possibility of anxiety or depression. HADS has been validated in its Swedish translation [31].

The Insomnia Severity Index (ISI) is a reliable and valid instrument for detecting cases of insomnia and has excellent internal consistency [32]. The seven items of the ISI are rated on a five-point Likert scale (0–4). The seven scores are summed to create the total ISI score (maximum 28).

The RAND 36-mental component summary (MCS) is one of the generic profile HRQoL measures used to compare the relative burden of chronic disease [32, 33]. Comprising 36 items, RAND-36 assesses eight health dimensions with multi-item scales. Further calculation yields eight scale scores (range 0–100). Two summary scores, physical and mental health composites (PCS and MCS), were also derived from these eight scales. We used MCS in this study.

Two Multidimensional Pain Inventory (MPI) subscales–affective distress (MPI-distress, 0 = no distress and 6 = very distressed) and perceived life control (MPI-LifeCon, 0 = poor control and 6 = good control)–were chosen. MPI is a self-report instrument that assesses psychosocial, cognitive, and behavioural effects of chronic pain using 61 items (34 items in the Swedish version MPI-S) on a scale ranging from 0 to 6 [34, 35]. The selected subscales reflect emotional functioning and the ability to cope with psychological distress [27, 36].

## Physical activity aspects

The Swedish National Board of Health and Welfare recommends three PA questions [37], which are included in the registry. This study includes two of these questions:

1. During a regular week, how much time do you spend exercising on a level that makes you short winded, for example, running, fitness class, or ball games? The following answer alternatives were provided: 0 minutes/none, less than 30 minutes, 30–60 minutes (0.5–1 hour), 60–90 minutes (1–1.5 hours), 90–120 minutes (1.5–2 hours), and more than 120 minutes (2 hours). A scale ranging from 1–6 (0 minute/none = 1 and more than 120 minutes = 6) was applied for this physical exercise variable and was denoted as PE.

2. During a regular week, how much time are you physically active in ways that are not exercise, for example, walks, bicycling, or gardening? Added together, all activities should last for at least 10 minutes. The following answer alternatives were provided: 0 minutes/none, less than 30 minutes, 30–60 minutes (0.5–1 hour), 60–90 minutes (1–1.5 hours), 90–150 minutes (1.5–2.5 hours), 150–300 minutes (2.5–5 hours), and more than 300 minutes (5 hours). Hence, the scale–denoted as EPA–ranged between 1 (0 minutes/none) and 7 (more than 300 minutes).

The first question refers to physical exercise (PE) and the second one to everyday physical activity (EPA). These categories were found to have the strongest validity comparison to open-end questions or to the time spent on daily PA [38]. Following the national recommendations [37], we also generated an outcome of PA volume by converting categorical options to activity minutes, denoted as PA time. The midpoints of intervals in each given answer option (i.e., less than 30 minutes converted to 15 minutes, 30–60 minutes converted to 45 minutes, more than 120 minutes or 300 minutes converted to 120 and 300 minutes, respectively) were used. The PA time was calculated by multiplying PE by two and adding the product to EPA (PE

minutes × 2 + EPA minutes). Less than 150 minutes per week indicate insufficient physical activity (in the following denoted as insufficient PA time) [37, 39]. The questionnaire has been validated for use in Sweden [40].

## Statistics

Traditional statistical analyses were performed with SPSS Statistics (IBM Corporation, Somers, NY, version 26.0). Mean with standard deviations (Mean ± SD) or median with interquartile range for continuous variables and number with percentage (n, %) for categorical variables were used to report descriptive data. The criteria for testing normality was $\geq \pm 2.00$ for the skewness and $\geq \pm 7.00$ for the kurtosis, since the typical use of the Kolmogorov-Smirnov and the Shapiro-Wilk tests is not recommended for large sample sizes [41]. The chi-square test was used to compare values of the categorical variables, and one-way ANOVA (Bonferroni method for post hoc test) was used for continuous data fulfilled both normality and homogeneity of variance. Kruskal-Wallis test and the Mann-Whitney U test with Bonferroni method for post hoc test were used when the assumption of homogeneity of variance among the groups were rejected by Levene's test ($P<0.05$). A $P$-value below 0.05 was regarded as significant. A $P$-value of less than 0.017 was used for Mann-Whitney U test, which served as a post hoc test to control for risk of mass significance [42]. Logistic regression (Likelihood ratio method) was used to determine the dichotomous parameter–i.e., sufficient/insufficient PA time. We also defined PA time as a tetrachotomous variable ($1^{st}$-$4^{th}$ quartile) and ordinal regression was subsequently used. Covariates were removed from the models if $P$-value was less than 0.1 from the prior model. Pearson's chi-square test (goodness of fit) and test of parallel lines (assumption of proportional odds) were calculated so that the models were not violated ($P > 0.05$). Multicollinearity was assessed by examining tolerance and the variance inflation factor (VIF); a VIF less than 2.5 might also indicate a multicollinearity problem [43].

The logistic regressions suffer some disadvantages, such as assumptions of variable independence, meeting power, and missing data [44–46]. Therefore, we also performed confirmatory analyses using advanced Principal Component Analysis (PCA) for the multivariate correlation analyses to detect outliers and Orthogonal Partial Least Square Regressions (OPLS) for the multivariate regressions. Analyses were conducted using SIMCA-P+ (version 15, Umetrics, Sartorius Stedim Biotech, Umeå). The confirmatory analyses were used in a larger proportion of the whole study sample because SIMCA-P+ uses the Nonlinear Iterative Partial Least Squares algorithm (NIPALS algorithm) to compensate for missing data. Detailed information about this statistical method is described in S1 Appendix.

We investigated the relative importance of BMI, socio-demographic characteristics, pain, and psychological aspects (X-variables) for insufficient PA and PA time quartiles (Y-variables) using OPLS. The importance of the X-variables was measured as a Variable Influence on Projection (VIP) value. VIP indicates the relevance of each X-variable pooled over all dimensions and Y-variables–i.e., the group of variables that best explain Y. VIP $\geq 1.0$ was considered significant if the VIP value had a 95% jack-knife uncertainty confidence interval non-equal to zero [45]. P(corr) was used to note the direction of the relationship (positive or negative). P(corr) depicts the loading of each variable scaled as a correlation coefficient, standardizing the range from -1 to +1. An absolute P(corr) $> 0.4$–0.5 is generally considered significant [47]. For each regression, we report the $R^2$, $Q^2$, and the result (i.e., $P$-value) of a cross-validated analysis of variance (CV-ANOVA). In addition, we required significant CV-ANOVA for a regression to be significant. A certain variable was considered significant when VIP $> 1.0$ and absolute p(corr) $> 0.40$.

## Results

### Missing data and background characteristics

A total of 3615 patients were registered in the SQRP when this study was conducted. Missing observations in the selected variables are listed in Fig 1. Compared to the patients included in the study (n = 3110), non-participants due to missing BMI information (n = 505) were more likely to be women (405/505, 80.2%), without a university/college education (406/505, 80.4%), and not currently working or studying (322/505, 63.8%). However, no statistical significance was found for age, place of birth, pain, and psychological aspects.

The logistic regression models revealed some demographic differences between excluded cases (non-participants and dropouts, n = 1316) and included cases (n = 2299). Compared to the included patients, the excluded patients were older, more likely to be born outside Europe, and more likely not working/studying. Fewer excluded patients reported constant pain ($P < 0.001$). The excluded patients reported worse mental health status (Rand-36 MCS, $P = 0.019$) and less PA time ($P = 0.019$). No statistical difference was found for absolute BMI or BMI groups ($P > 0.05$) (see S1 Table).

A majority of the 3110 patients (44.5±12.2 years) were women (74.8%), born in Europe (86.9%), and without university/college education (72.7%). About half of the patients were

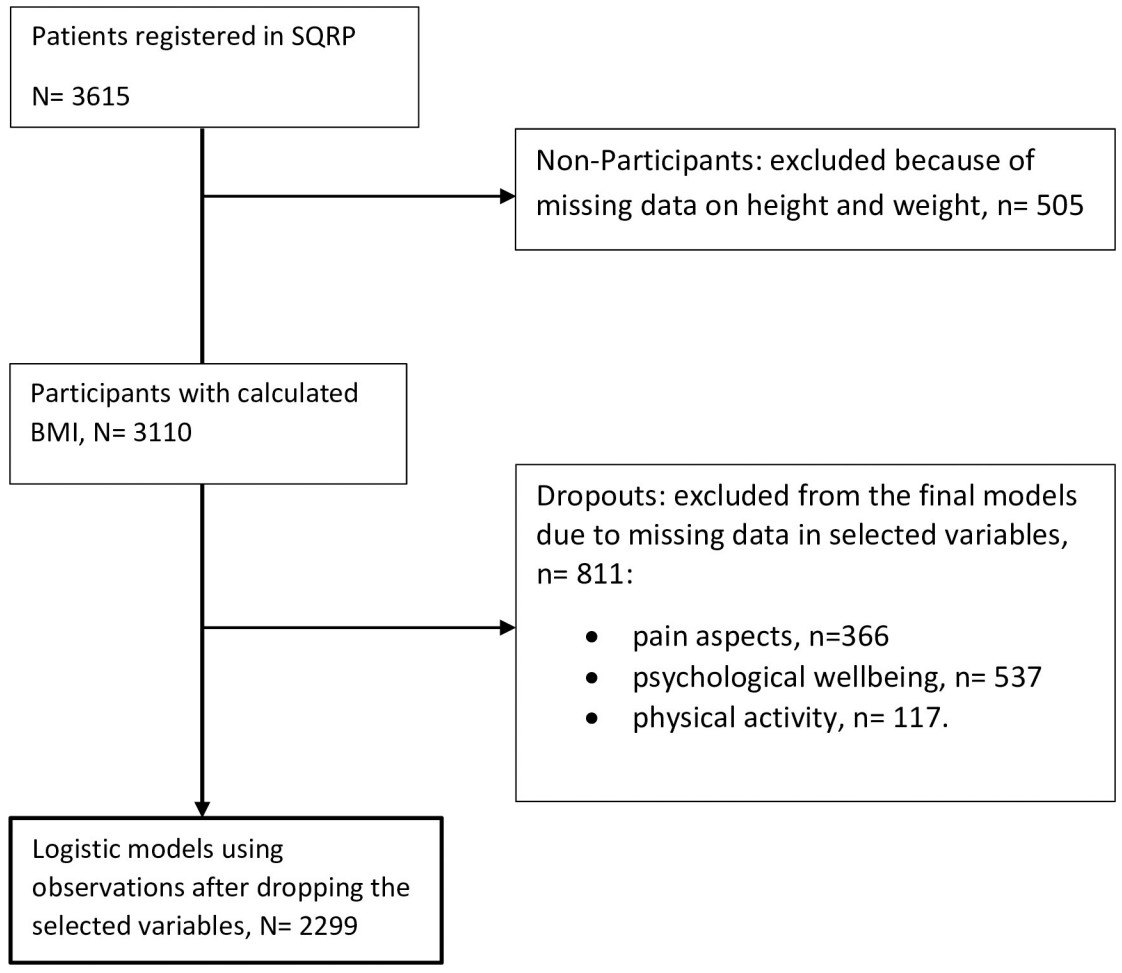

**Fig 1. Flow chart outline of the inclusion of participants.**

currently working or studying (51.2%). One-fourth of the patients were obese (871/3110, 25.3%) and nearly one-third of these obese patients were severely obese (242/871, 27.8%). Very few were categorized in the underweight group (1.5%). Some differences were found between the BMI groups: patients in the underweight group were younger and more likely working/ studying; patients born outside Europe were slightly more common in the overweight and mild obese groups; and patients with severe obesity were more likely to be women, have lower education, and not currently working/studying. Although statistically significant, these differences were clinically and numerically small (Table 1).

## Pain and psychological aspects

Pain characteristics and psychological symptoms in the different BMI groups are shown in Table 1. Obese patients had in general a worse pain profile than the normal weight patients

**Table 1. General characteristics, N = 3110.**

| | Total | Underweight | Normal weight | Overweight | Mild obesity | Severe obesity | *P*-values |
|---|---|---|---|---|---|---|---|
| **BMI, Mean±SD** | 27.1±5.4 | 17.7±0.9 | 22.4±1.7 | 27.4±1.4 | 32.0±1.4 | 39.3±4.1 | - |
| **BMI category, n (%)** | 3110 | 46 (1.5%) | 1166 (37.5) | 1113 (35.8) | 543 (17.5) | 242 (7.8) | - |
| **Socio-demographic factors** | | | | | | | |
| **Women, n (%)** | 2327 (74.8) | 40 (87.0) | 936 (80.3) | 756 (67.9) | 392 (72.2) | 203 (83.9) | <0.001 |
| **Age, years, Mean±SD** | 44.5±12.2 | 36.5±13.5 | 42.2±12.9 | 45.5±11.8 | 47.2±11.2 | 46.7±9.9 | <0.001 |
| **Born outside Europe, n (%)** | 406 (13.1) | 5 (10.9) | 114 (9.8) | 180 (16.2) | 82 (15.1) | 25 (10.3) | <0.001 |
| **University/college, n (%)** | 835 (27.3) | 12 (26.1) | 393 (33.7) | 270 (24.3) | 126 (23.2) | 46 (19) | <0.001 |
| **Work/study, n (%)** | 1592 (51.2) | 28 (60.9) | 630 (54.0) | 575 (51.7) | 249 (45.9) | 110 (45.5) | 0.005 |
| **Pain aspects [1]** | | | | | | | |
| **Pain intensity (NRS-7days), Mean±SD** | 6.9±1.9 | 7.4±1.2 | 6.6±1.9 | 7.1±1.7 | 7.1±1.8 | 7.4±1.5 | <0.001 [ab, bd, be] |
| **Pain distribution (PRI), Mean±SD** | 15±9 | 18±9 | 14±8 | 14±8 | 16±9 | 18±9 | <0.001 [ab, bd, be] |
| **Pain-duration, months, median (q1-q3)** | 160 (66–405) | 167 (66–370) | 124 (59–335) | 167 (67–400) | 207 (75–500) | 200 (93–492) | <0.001 [ab, bd, be] |
| **Presence of constant pain, n (%)** | 2482 (79.8) | 37 (80.4) | 861 (73.8) | 913 (82.0) | 462 (85.1) | 209 (86.4) | <0.001 |
| **Psychological aspects [2]** | | | | | | | |
| **HADS-A, Mean±SD** | 9.4±4.7 | 10.1±5.5 | 9.5±4.7 | 9.2±4.7 | 9.4±4.7 | 9.3±4.9 | NS |
| **HADS-D, Mean±SD** | 9.1±4.5 | 9.3±4.3 | 8.9±4.6 | 9.0±4.5 | 9.6±4.4 | 9.7±4.4 | 0.016 [bd] |
| **Insomnia index, Mean±SD** | 16.7±6.6 | 16.5±6.4 | 15.6±6.7 | 17.0±6.6 | 17.4±6.5 | 18.5±6.3 | <0.001 [bd, bc, be] |
| **RAND-36 MCS, Mean±SD** | 41.0±21.4 | 37.3±20.5 | 41.5±21.5 | 41.3±21.9 | 39.5±20.4 | 40.3±21.5 | NS |
| **MPI-affective distress, Mean±SD** | 3.5±1.3 | 3.6±1.4 | 3.5±1.3 | 3.5±1.3 | 3.5±1.3 | 3.5±1.4 | NS |
| **MPI-lifecon, Mean±SD** | 2.7±1.2 | 2.5±1.1 | 2.7±1.2 | 2.8±1.2 | 2.6±1.2 | 2.6±1.2 | NS |
| **Physical activity (PA) [3]** | | | | | | | |
| **PA time /week, minutes, Median (q1-q3)** | 150 (75–300) | 150 (75–300) | 210 (75–315) | 165 (75–300) | 120 (45–225) | 120 (45–225) | <0.001 [bd, bc, be] |
| 1st quartile (0-75min), n (%) | 957 (32) | 12 (27.9) | 308 (27.2) | 331 (31.1) | 206 (40.0) | 100 (42.2) | <0.001 |
| 2nd quartile (>75-150min), n (%) | 543 (18.1) | 10 (23.3) | 182 (16.1) | 195 (18.3) | 102 (19.8) | 54 (22.8) | – |
| 3rd quartile (>150-300min), n (%) | 844 (28.2) | 14 (32.6) | 330 (29.2) | 328 (30.8) | 120 (23.3) | 52 (21.9) | – |
| 4th quartile (>300 min), n (%) | 649 (21.7) | 7 (16.3) | 312 (27.6) | 212 (19.9) | 87 (16.9) | 31 (13.1) | – |
| **Insufficient PA, n (%)** | 1396 (47.3) | 18 (41.9) | 460 (40.6) | 498 (46.7) | 296 (57.5) | 142 (59.9) | <0.001 |

Chi-square, one-way ANOVA (Post-hoc test with Bonferroni) or Kruskal–Wallis test (Mann-Whitney U test with Bonferroni method for post hoc test).

a: underweight group = 366

b: normal weight group

c: overweight group

d: mild obesity group

e: severe obesity group

Missing data: [1] = 366, [2] = 537, [3] = 117. NS: not significant.

(e.g., significantly higher intensity, more spreading pain (i.e., higher PRI), and longer pain duration. Most patients described their pain as being constant (79.8%), fewer in in the normal weight group (73.8%) and more in the severely obese group (86.1%, $P < 0.001$).

There were no significant differences regarding psychological aspects across BMI groups other than insomnia and depression. Both overweight and obese patients reported levels of insomnia higher than the normal weight group (ISI; $P < 0.001$). Compared to the other groups, the mild obese group had marginally higher depressive symptoms (HADS-D).

## Physical activity

There was a large variation with regard to PE across the BMI groups (Fig 2). Over 40% of patients in each group (42.9–51.4%, P = 0.014, $\chi2$ = 12.58) reported not participating in any physical exercise. Patients in the obese groups and underweight group performed significantly less PE than normal weight patients in each accumulated scale (cut-offs: 30 mins, 60 mins, 90 mins and 120 mins; $P < 0.001$, $\chi2$ = 36.11–42.96; odds ratios are given in S1 Fig).

Analyses of EPA (Fig 3) showed a similar pattern: higher BMI was associated with less EPA. A clear trend was evident for the category 30–60 min /week and above (cut-offs: 60 mins, 90 mins, 150 mins and 300 mins): patients with higher BMI spent less time on EPA ($P < 0.001$, $\chi2$ = 26.86–41.95; odds ratios in S2 Fig).

Overall, obese patients had significantly less PA time (median: 120 min/week) than normal weight patients (median time 210 min/week, $P < 0.001$) (Table 1). PA time, distributed in quartiles, revealed that there were fewer patients with severe obesity (13.1%) (i.e., had less hits) in the highest PA quartile (>300 min/week). Nearly half of the patients were classified as having insufficient PA (47.3%) (Table 1); severe obesity had the highest proportion (59.9%), while normal weight had the lowest (40.6%, $P < 0.001$).

Binary logistic regression showed that patients in the higher BMI categories were more likely to report insufficient PA time: overweight patients had a 28% increased risk and severe obese patients had a 118% increased risk (Table 2; Model 1). After adjusting for socio-

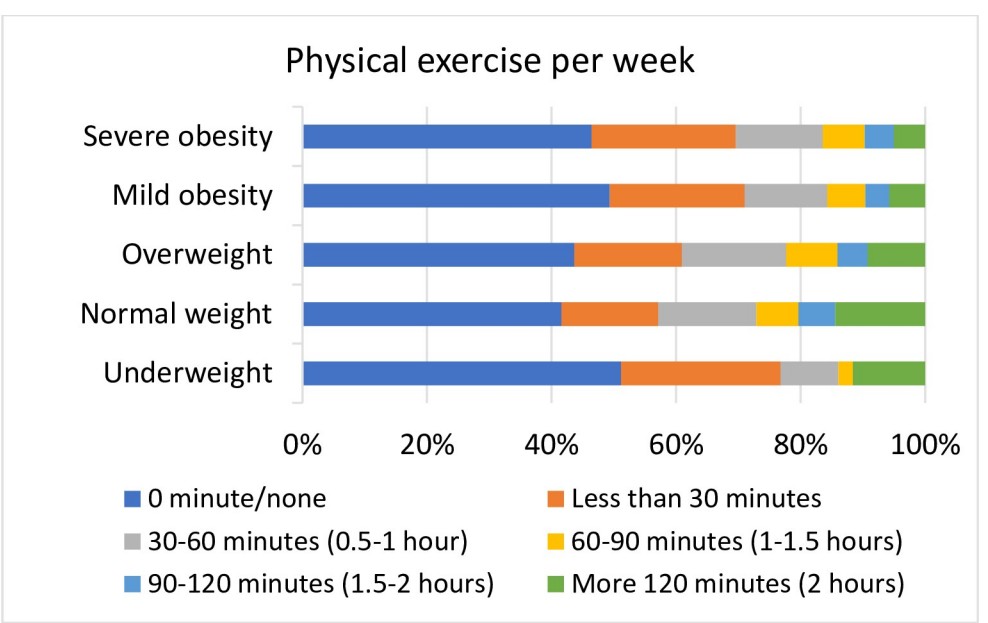

**Fig 2. Physical exercises Per Week (PE) in the different BMI categories.**

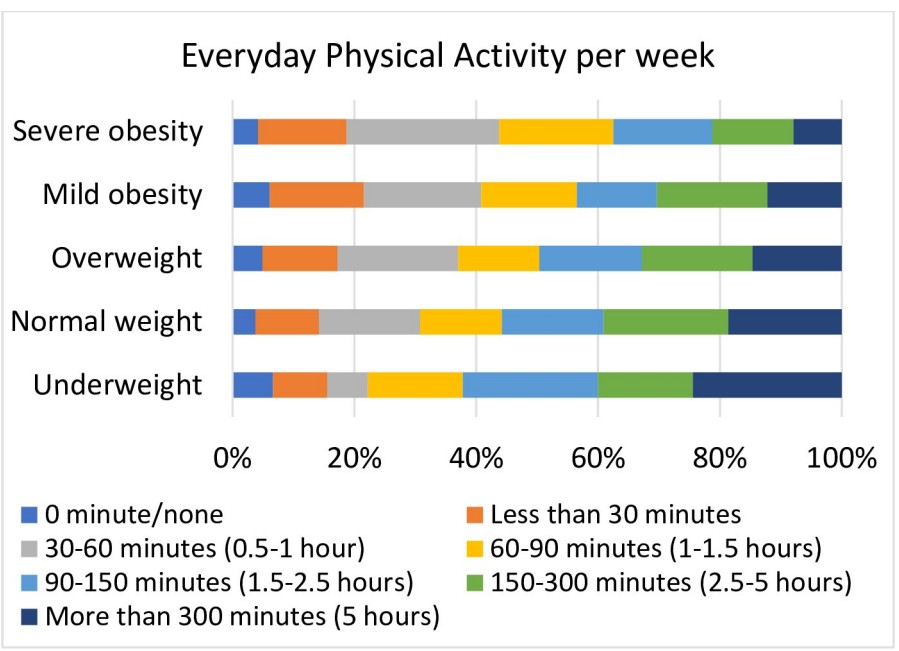

**Fig 3. Everyday Physical Activity per week (EPA) in the different BMI categories.**

demographic factors, pain, and psychological aspects, the odds remained relatively robust for the two obesity categories (mild obesity: OR 1.65, 95% CI 1.29–2.12, severe obesity: OR 1.96, 95% CI 1.41–2.73) (Table 2; Model 2). Being born outside Europe (OR 1.81, 95% CI 1.37–2.39) was a strong predictor of insufficient PA. Pain variables (e.g., pain intensity and pain spreading) and variables related to psychological distress (e.g., HADS-anxiety and HADS-depression) had relatively weak although significant prediction value. When stratified by obesity category (BMI $\geq$ 30 kg/m$^2$), only pain intensity (OR 1.17, 95% CI 1.06–1.29) was positively associated with insufficient PA time.

Using ordinal regression, we found that higher BMI categories (from the overweight level and above) were negatively associated with PA time (Table 3; Model 1). Excess weight at the obesity level (mild or severe obesity) had a significant reverse effect on more PA time when other factors were entered (Table 3; Models 2, 3, and 4; OR 0.51–0.65). Some socio-demographic factors contributed significantly to the model, but the contributions weakened as variables of pain and psychological aspects were entered. Two variables–born outside Europe and having university/college education–remained significant for PA time although in different directions (Model 4). University/college education level (OR 1.98, 95% CI 1.37–2.8) kept the positive association even after stratified by obesity (Model 5). Although several parameters in pain aspects and psychological wellbeing showed statistical significance in the models, the effects on increasing PA time were considered negligible or at best limited (OR and 95% CI closed to 1, Models 3–5).

Over 3200 patients were included in the confirmatory analysis as it was possible to handle missing data to a certain level. The confirmatory analysis agreed with the logistic analysis in that BMI and pain intensity were important for PA level (Table 4). Unlike the analyses in Table 2 (Model 2) and Table 3 (Model 4), the analysis in Table 4 revealed that insomnia was a significant regressor with a negative influence on PA. The OPLS regressions confirmed HADS-depression but not HADS-anxiety as significant. As in the analysis in Table 3, the OPLS regressions confirmed that certain sociodemographic characteristics influenced PA.

**Table 2. Logistic regression of factors related to insufficient PA time.**

|  | Model 1: BMI category | Model 2: adjusted | Model 3: Stratified by obesity |
|---|---|---|---|
| **BMI category (reference: normal weight)** |  |  | NA |
| Underweight | 1.05 (0.57–1.95) | 1.09 (0.50–2.34) | NA |
| Overweight | 1.28 (1.09–1.52)** | 1.15 (0.94–1.41) | NA |
| Mild obesity | 1.98 (1.60–2.44)*** | 1.65 (1.29–2.12)*** | NA |
| Severe obesity | 2.18 (1.64–2.91) *** | 1.96 (1.41–2.73)*** | NA |
| **Sex (women = 1)** |  | NS | NS |
| **Age, years** |  | 1.01 (1.00–1.02)* | NS |
| **Born outside Europe (yes = 1)** |  | 1.81 (1.37–2.39)*** | NS |
| **University/college (yes = 1)** |  | 0.77 (0.63–0.94)* | NS |
| **Working/Study (yes = 1)** |  | NS | NS |
| **Pain intensity (NRS-7 days)** |  | 1.10 (1.04–1.16)* | 1.17 (1.06–1.29)** |
| **Pain distribution (PRI)** |  | 1.02 (1.00–1.03)* | NS |
| **Pain-duration, months** |  | NS | NS |
| **Constant pain (yes = 1)** |  | NS | NS |
| **HADS-anxiety** |  | 0.96 (0.94–0.99)** | NS |
| **HADS-depression** |  | 1.06 (1.03–1.09)** | NS |
| **Insomnia index** |  | NS | NS |
| **RAND-36 MCS** |  | NS | NS |
| **MPI-Distress** |  | NS | NS |
| **MPI-LifeCon** |  | 0.88 (0.80–0.96)* | NS |
| **Nagelkerke $R^2$** | 0.025 | 0.095 | 0.021 |

NS: not significant. NA: not applicable.

* $P < 0.05$

** $P < 0.01$

*** $P < 0.001$.

## Discussion

This study demonstrates that patients referred to specialist pain rehabilitation clinics were less likely to participate in sufficient PA. In addition to non-malignant chronic pain, being obese was associated with low self-reported PA levels. As the obesity epidemic is spreading fast, it is important to attend to both higher BMI and lack of PA. To this end, this study uses a wide range of variables from a large cohort in a national data registry. Patients with chronic pain and comorbid obesity can be helped to increase PA, a shared goal in pain rehabilitation, by enhancing our understanding of the factors associated with low PA.

### Physical activity among patients with chronic pain reporting different BMI

About 50% of the patients in this study population reported insufficient PA in relation to recommendations from the Swedish National Board of Health and Welfare. This estimate is larger than that of the general Swedish population, reportedly with a prevalence of about 34–35% (data source in 2016 and 2018) [48]. We might be underestimating the problem due to the nature of self-reported data. The estimates are even larger in studies that use objective PA measurements such as studies of fibromyalgia patients [49]. Interestingly, this study found a variation of reported PA, including PE and EPA, across the BMI groups. Regular performance of PE depends not only on the individuals' abilities and limitations, but also on factors related to lifestyles and goals in adulthood. Furthermore, reported EPA decreased gradually as higher

**Table 3. Determinants of PA time (in quartiles): Ordinal regression.**

| | Model 1, BMI category | Model 2, adjusted for socio-demographic factors | Model 3, adjusted for chronic pain profiles | Model 4, adjusted for psychological well-being | Model 5 Stratified by obesity |
|---|---|---|---|---|---|
| **BMI category (reference: normal weight)** | | | | | |
| Underweight | 0.73 (0.42–1.26) NS | 0.71 (0.41–1.24) NS | 0.79 (0.42–1.49) NS | 0.76 (0.40–1.44) NS | NA |
| Overweight | 0.75 (0.65–0.88) *** | 0.83 (0.71–0.97) * | 0.86 (0.73–1.01) NS | 0.84 (0.71–1.00) NS | NA |
| Mild obesity | 0.53 (0.44–0.64) *** | 0.60 (0.49–0.72) *** | 0.64 (0.52–0.79) *** | 0.65 (0.53–0.81) *** | NA |
| Severe obesity | 0.45 (0.35–0.58) *** | 0.51 (0.39–0.66) *** | 0.54 (0.41–0.71) *** | 0.56 (0.42–0.74) *** | NA |
| **Sex (reference: men)** | | 0.89 (0.77–1.04) NS | NA | NA | NS |
| **Age, years** | | 0.99 (0.99–1.0) ** | 0.99 (0.99–1.00) ** | 0.99 (0.99–1.00) * | 0.98 (0.97–1.00) * |
| **Born outside Europe** | | 0.55 (0.45–0.67) *** | 0.64 (0.51–0.79) *** | 0.60 (0.47–0.76) *** | NS |
| **University/college** | | 1.49 (1.29–1.73) *** | 1.34 (1.14–1.58) *** | 1.37 (1.16–1.61) *** | 1.98 (1.37–2.87) *** |
| **Working/Study** | | 1.36 (1.20–1.56) *** | 1.30 (1.13–1.50) *** | 1.13 (0.97–1.32) NS | NS |
| **Pain intensity (NRS-7 days)** | | | 0.92 (0.88–0.96) *** | 0.94 (0.90–0.99) * | 0.88 (0.79–0.98) * |
| **Pain distribution (PRI)** | | | 0.99 (0.98–1.00) ** | 0.98 (0.98–0.99) ** | NS |
| **Pain-duration, months** | | | 1.0 (1.0–1.0) NS | NA | NS |
| **Constant pain (yes = 1)** | | | 1.02 (0.84–1.23) NS | NA | NS |
| **HADS-anxiety** | | | | 1.05 (1.03–1.08) *** | 1.08 (1.03–1.13) ** |
| **HADS-depression** | | | | 0.95 (0.93–0.98) *** | 0.92 (0.88–0.97) ** |
| **Insomnia index (ISI)** | | | | 1.0 (0.99–1.02) NS | 1.03 (1.01–1.06) * |
| **RAND-36 MCS** | | | | 1.0 (1.00–1.01) NS | NS |
| **MPI-Distress** | | | | 0.94 (0.86–1.03) NS | NS |
| **MPI-LifeCon** | | | | 1.16 (1.06–1.26) ** | NS |
| **Nagelkerke R2** | 0.02 | 0.06 | 0.07 | 0.09 | 0.09 |

NS: not significant. NA: not applicable

* $P < 0.05$

** $P < 0.01$

*** $P < 0.001$.

Variables removed (NA) in Model 2–4: covariates were removed from the current model if $P$ value >0.1 from the prior model; variable removed (NA) in Model 5: stratification by BMI $\geq 30 kg/m^2$.

BMI categories were considered. These findings agree with reported links between non-exercise activity thermogenesis (NEAT) and incidence of excess weight, especially obesity-related comorbidities [50, 51]. Our results from a chronic pain patient population indicate that obese pain patients face negative consequences due to excess weight (e.g., low function capacity, activity limitation, and comorbidities) not experienced by non-obese pain patients [14, 16].

## Association between physical activity and BMI in patients with chronic pain

In line with studies that focus on specific pain conditions [52–54], higher BMI categories were associated with less PA time. Obese patients were more likely to be below the level of sufficient PA time and the effect of obesity on PA was relatively strong (Table 2; Model 2). Obesity modulates pain via mechanical loading [55], proinflammatory cytokines and other substances [56, 57], and psychological strain [58, 59]. Therefore, obese patient with insufficient PA level will

**Table 4. Regressions of insufficient PA time (OPLS-DA) and PA time in quartiles (OPLS).**

| Dependent: | Insufficient PA time | | Dependent: | PA time (in quartiles) | |
|---|---|---|---|---|---|
| **Regressors** | **VIP** | **p(corr)** | **Regressors** | **VIP** | **p(corr)** |
| **Pain intensity** | **1.61** | **0.63** | **Pain intensity** | **1.52** | **-0.61** |
| **BMI** | **1.34** | **0.53** | **BMI** | **1.33** | **-0.53** |
| **Insomnia index** | **1.28** | **0.50** | **Insomnia index** | **1.22** | **-0.48** |
| **University/college** | **1.11** | **-0.43** | **HADS-depression** | **1.14** | **-0.46** |
| **Born outside Europe** | **1.06** | **0.42** | **MPI-Lifecon** | **1.13** | **0.45** |
| **HADS-depression** | **1.06** | **0.42** | **RAND-36 MCS** | **1.12** | **0.44** |
| **RAND-36 MCS** | **1.04** | **-0.40** | **University college** | **1.12** | **0.44** |
| **MPI-Lifecon** | **1.04** | **-0.41** | **Working/study** | **1.03** | **0.41** |
| **Working/study** | **1.03** | **-0.41** | Born outside Europe | 0.97 | -0.39 |
| Age | 0.95 | 0.37 | MPI-distress | 0.90 | -0.36 |
| Pain distribution (PRI) | 0.88 | 0.34 | Pain distribution (PRI) | 0.90 | -0.36 |
| MPI-distress | 0.84 | 0.33 | Age | 0.89 | -0.36 |
| constant pain | 0.69 | -0.27 | constant pain | 0.67 | 0.27 |
| HADS anxiety | 0.58 | 0.23 | HADS-anxiety | 0.60 | -0.24 |
| Pain duration | 0.25 | 0.10 | Pain duration | 0.34 | -0.14 |
| Sex (woman) | 0.00 | 0.00 | Sex (woman) | 0.04 | 0.01 |
| $R^2$ | 0.07 | | $R^2$ | 0.08 | |
| $Q^2$ | 0.06 | | $Q^2$ | 0.08 | |
| CV-ANOVA p-value | <0.001 | | CV-ANOVA p-value | <0.001 | |
| n | 3 277 | | n | 3 277 | |

Notes: BMI was arranged as an ordinal variable (0 = underweight to 4 = Obesity II-III).

Abbreviations: VIP (very important projection, VIP>1.0 is significant), corr (absolute p(corr) >0.40 is significant) are reported. The sign of p(corr) indicates the direction of the correlation with the dependent variable (+ = positive correlation;— = negative correlation). The four bottom rows of each regression report $R^2$, Q2, and P-value of the CV-ANOVA and number of patients included in the regression (n).

not gain the benefits of PA, such as weight control [51], inflammation reduction [23], and better psychological function [20].

In the logistic regressions, pain and psychological aspects showed a very limited explanatory value for PA time. PA is a voluntary action affected by many factors, including socio-demographic background and environmental influences as well as pain and psychological aspects. Our results suggest that the association between excess weight and PA was partly due to the contribution of these factors. Although pain, psychological features, and obesity can influence PA behaviour, none of these factors were dominant due to potential interactions. The advanced statistical analyses (OPLS) provided more information when interrelationships between covariates and missing data were properly handled. BMI, pain intensity, and the insomnia index appeared as the three most important factors related to insufficient PA and PA time. Furthermore, Rand-36 MCS and HADS-depression (not HADS-anxiety) had significant explanation value for variations in PA time. Although obvious multicollinearity was not identified by examining tolerance and the variance inflation factor (VIF), the potential interrelationships between pain, obesity, and psychological aspects could still affect the independence of covariates in logistic regressions, especially as their effects tended to have a modest or close to insignificant borderline influence. In our previous research, pain intensity, anxiety, and depression were important regressors of Rand-36 MCS and had weak correlation with insomnia [60]. When obesity was stratified, the only factor associated with insufficient PA time was pain intensity (Table 2; Model 2); the insomnia index showed only a marginal impact on PA

time (Table 3; Model 5). These findings revealed to some extent their interactions with obesity. Importantly, although we obtained significant logistic and ordinal regressions (Tables 2 and 3) as well as significant OPLS regressions (Table 4), the explained variations in the investigated PA variables were below 10% (e.g., OPLS regressions: $R^2$ = 0.07–0.08). Hence, we have an incomplete understanding of factors that determine the PA levels in patients with chronic pain.

## Optimal interventions for chronic pain patients with obesity (clinical implications)

The health initiative 'Exercise is Medicine' of the American College of Sports Medicine (ACSM) [61] and recent studies [20, 24, 62] conclude that PA has an important role in pain management and rehabilitation. Based on a biopsychosocial model in pain rehabilitation [63], supervised PA (including PE performance) involves both physiological actions (biological) and lifestyle behaviour (psychological). As PA is a modifiable factor at the individual level, PA interventions should consider the individual's baseline levels. To date, there is no well-established PA program that applies to patients with complex chronic pain conditions [20, 24]. Some studies have discussed which type of PA best helps pain patients with complex problems. For example, Häuser et al. report that regular aerobic exercise programs should contain exercise slight to moderate intensity [64]. Exercise-induced hypoalgesia as well as hyperalgesia were found in patients with different chronic pain syndromes [65]. Difficulties with and failure to perform PA have been reported in qualitative studies, even though the patients with chronic pain were aware of the positive effect of PA and highly valued PA as a treatment [66, 67].

Health professionals should find viable interventions in relation to their patients' complex problems. In a recent study, we reported that severe obese patients reported the least improvement after participating in an IMMRP [68]. In Sweden, IMMRPs generally contain education, cognitive behavioural therapy, physical exercise, and planning and adaptations in order to return to work or studies [36]. Although most IMMRPs are group-based, individualised adaptations should be emphasised. As at least 50% of chronic pain patients being treated at the specialist level have sleep problems (e.g., insomnia), we have suggested that IMMRPs should also offer insomnia treatment. Similarly, multimodal treatment portfolios for obese patients should include special interventions to increase PA level and reduce BMI. Reasonably, these interventions might start by modifying EPA for patients with excess weight. This conclusion is supported by previous literature that found that increasing and accumulating short bouts of daily activity (walking, using the stairs, gardening, etc.) help obese patients increase their PA [50, 69].

## Methodological considerations, strengths, sand limitations

Fewer than 65% of our sample (i.e., patients recorded in the SQRP) had a complete data set. The statistical analysis of the descriptive data was performed to reveal the representation of the current sample in the final logistic regression models. No robust multicollinearity was detected prior to logistical regressions. However, it is plausible to consider confirmatory analyses because the risks for intercorrelations among the covariates still could exist. OPLS regressions have higher sensitivity, properly deal with multicollinearity, compensate for missing data, and provide higher statistical power [45, 46, 70]. However, OPLS cannot determine the independent effect of a certain variable when controlling for other variables. A strength of this study is its use of advanced statistical methods when several variables with borderline significance were found using classic regression analysis. In addition, both the advanced and the classic statistical methods revealed that obesity was a strong predictor for insufficient PA and less PA time.

Moreover, there are certainly some limitations beyond the cross-sectional nature of the comparisons. First, data used in this study were collected at one time point, so no causal relationship could be explored or concluded. Second, self-reported PA variables have limited validity and reliability as they can both overestimate or underestimate PA [71]. Nonetheless, unlike artificial laboratory settings, self-reported PA is a common and efficient way to measure daily PA. Moreover, self-reported PA is especially useful when investigating a large clinical patient population. The two questions about PA are in line with other more extensive PA questionnaires, and the Swedish National Board of Health and Welfare recommends that these questions be used as a screening tool in clinical practice [37, 38]. Third, the present study used a sample from a Swedish population. Questions about PA behaviour included in this questionnaire did not cover other lifestyle aspects. This might limit generalizability as many factors, especially cultural and environmental factors, influence people's lifestyles, including PA [72]. Future studies in pain and rehabilitation should consider these factors and clinicians should consider their patients' specific backgrounds and environmental conditions when modifying PA interventions.

## Conclusion

Typically, patients referred to pain rehabilitation clinics had low PA levels and this is especially true for patients with comorbid obesity. Clearly, efforts should be made to encourage obese patients to be more physically active. The first step might be to increase slowly the intensity and amount of less painful daily PA.

## Supporting information

**S1 Appendix. Orthogonal Partial Least Square Regressions (OPLS) statistical method.** (DOCX)

**S1 Table. Characteristics of patients included in logistic regression (N = 2299) and excluded cases (missing BMI n = 505, and dropouts n = 811).** (DOCX)

**S1 Fig. Likelihood (OR and 95% CI) of patients in different BMI categories (reference: normal weight) performing physical exercise (PE) per week.** (TIF)

**S2 Fig. Likelihood (OR and 95% CI) of patients in different BMI categories (reference: normal weight) taking Everyday Physical Activity (EPA) per week.** (TIF)

## Acknowledgments

We are very grateful to Annelie Inghilesi Larsson Quality Stat AB for extracting the data from SQRP. Special thanks to our senior consultant psychiatrist Peter Alföldi for his advice on interpreting results related to clinical practice.

## Author Contributions

**Conceptualization:** Huan-Ji Dong, Britt Larsson, Björn Gerdle.

**Data curation:** Huan-Ji Dong, Björn Gerdle.

**Formal analysis:** Huan-Ji Dong, Björn Gerdle.

**Funding acquisition:** Huan-Ji Dong, Britt Larsson, Björn Gerdle.

**Investigation:** Marcelo Rivano Fischer.

**Methodology:** Huan-Ji Dong, Britt Larsson, Marcelo Rivano Fischer, Björn Gerdle.

**Project administration:** Marcelo Rivano Fischer.

**Software:** Björn Gerdle.

**Supervision:** Britt Larsson, Björn Gerdle.

**Writing – original draft:** Huan-Ji Dong, Björn Gerdle.

**Writing – review & editing:** Huan-Ji Dong, Britt Larsson, Marcelo Rivano Fischer, Björn Gerdle.

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
