## [Decision Letter · Decision Letter 0]

27 Jul 2020

PONE-D-19-34615

Facing obesity in pain rehabilitation clinics: profiles of physical activity in patients with chronic pain and obesity – a study from the Swedish Quality Registry for Pain Rehabilitation (SQRP)

PLOS ONE

Dear Dr. Dong,

Thank you for submitting your manuscript to PLOS ONE. After careful consideration, we feel that it has merit but does not fully meet PLOS ONE’s publication criteria as it currently stands. Therefore, we invite you to submit a revised version of the manuscript that addresses the points raised during the review process.

We look forward to receiving your revised manuscript.

Kind regards,

Adewale L. Oyeyemi, Ph.D

Academic Editor

PLOS ONE

Journal Requirements:

"This study was supported by grants from the County Council of Östergötland (Research-ALF) and the division research funding (SC-2017-00202-28). The funding source did not participate in the design of the study, collection, analysis, or interpretation of the data; or in the decision to submit the manuscript for publication.".

i) Please provide an amended statement that declares *all* the funding or sources of support (whether external or internal to your organization) received during this study, as detailed online in our guide for authors at http://journals.plos.org/plosone/s/submit-now.  Please also include the statement “There was no additional external funding received for this study.” in your updated Funding Statement.

ii) Please include your amended Funding Statement within your cover letter. We will change the online submission form on your behalf.

Reviewers' comments:

Reviewer's Responses to Questions

**Comments to the Author**

1. Is the manuscript technically sound, and do the data support the conclusions?

Reviewer #1: Yes

2. Has the statistical analysis been performed appropriately and rigorously? 

Reviewer #1: Yes

3. Have the authors made all data underlying the findings in their manuscript fully available?

Reviewer #1: Yes

4. Is the manuscript presented in an intelligible fashion and written in standard English?

Reviewer #1: Yes

5. Review Comments to the Author

Reviewer #1: The authors sought to investigate the levels of physical activity and the effect of excess weight on the levels this activity through analyses of data of patients attending specialist pain rehabilitation clinics. The manuscript is well written and presented appropriately. Rigourous statistical analysis techniques have been employed and are described in suitable detail for reproduction. Reasonable interpretations have been made from the results of the analyses and limitations of the work are clearly recognised. The article would benefit form the following minor revisions:

* Page 8 - Please change '....insufficient physical active' to activity.

* Table 1 - Percentage of women in underweight group is incorrect, should be 87.0.

* Page 14 - The statement 'most of these were severely obese patients (59.9%, P < 0.001)' is incorrect in this context. The percentage reported here is the proportion of the severe obesity group that had insufficient PA. The largest proportion of patients with insufficient PA came from the overweight (35.7%) group, with 10.2% of thetotal being in the severe obesity group.

* Page 19 - Please change '...being obese were' to 'being obese was'

6. PLOS authors have the option to publish the peer review history of their article (what does this mean?). If published, this will include your full peer review and any attached files.

Reviewer #1: No

---

## [Author Response · Author response to Decision Letter 0]

20 Aug 2020

Responses

Academic editor

Our comments and revisions: thank you for the detailed help on our article. In the revised manuscript, we have followed the guidelines and checked the title page, manuscript main body as well as supporting information (format and structures). We also used the recommended Preflight Analysis and Conversion Engine (PACE) digital diagnostic tool to ensure that figures meet PLOS requirements.

To be noted, we have updated our affiliations because Linköping University and Lund University changed a few names of affiliations and departments in January 2020. 

Our comments and revisions: Thank you for your clarification. We need to update our statement about data availability: The datasets generated and/or analysed in this study are not publicly available as the Ethical Review Board has not approved the public availability of raw data. The data that support the findings of this study are available from SQRP (https://www.ucr.uu.se/nrs/) but restrictions apply to the availability of these data, which were used under license for the current study, and so are not publicly available. Data are however available upon reasonable request and with permission of SQRP research group (contact person: Marcelo Rivano Fischer, Marcelo.rivanofischer@skane.se) and data requests should be sent to the Ethical Review Board.

"This study was supported by grants from the County Council of Östergötland (Research-ALF) and the division research funding (SC-2017-00202-28). The funding source did not participate in the design of the study, collection, analysis, or interpretation of the data; or in the decision to submit the manuscript for publication.".

i) Please provide an amended statement that declares *all* the funding or sources of support (whether external or internal to your organization) received during this study, as detailed online in our guide for authors at http://journals.plos.org/plosone/s/submit-now. Please also include the statement “There was no additional external funding received for this study.” in your updated Funding Statement.

ii) Please include your amended Funding Statement within your cover letter. We will change the online submission form on your behalf.

Our comments and revisions: Thank you for the information. We have completed some more necessary information according the PLoS guideline: This study was supported by grants from AFA insurance, the County Council of Östergötland (Research-ALF, LIO-608021, BG and SC-2017-00202-28, H-JD). AFA Insurance, a commercial founder, is owned by Sweden's labor market parties: The Confederation of Swedish Enterprise, the Swedish Trade Union Confederation (LO), and The Council for Negotiation and Co‐operation (PTK). These parties insure employees in the private sector, municipalities and county councils. The funding sources did not participate in the design of the study, collection, analysis, or interpretation of the data; or in the decision to submit the manuscript for publication. There was no additional external funding received for this study.

We have updated this funding statement in the resubmission.

Reviewer 1

Reviewer's Responses to Questions

Comments to the Author

1. Is the manuscript technically sound, and do the data support the conclusions?

Reviewer #1: Yes

2. Has the statistical analysis been performed appropriately and rigorously? 

Reviewer #1: Yes

3. Have the authors made all data underlying the findings in their manuscript fully available?

Reviewer #1: Yes

4. Is the manuscript presented in an intelligible fashion and written in standard English?

Reviewer #1: Yes

5. Review Comments to the Author

Reviewer #1: The authors sought to investigate the levels of physical activity and the effect of excess weight on the levels this activity through analyses of data of patients attending specialist pain rehabilitation clinics. The manuscript is well written and presented appropriately. Rigourous statistical analysis techniques have been employed and are described in suitable detail for reproduction. Reasonable interpretations have been made from the results of the analyses and limitations of the work are clearly recognised. The article would benefit form the following minor revisions:

* Page 8 - Please change '....insufficient physical active' to activity.

Our comments and revisions: Point taken. Thank you for your advice. We have changed the word in the text.

* Table 1 - Percentage of women in underweight group is incorrect, should be 87.0.

Our comments and revisions: Point taken. Thank you for your detailed help on our article. We have checked the calculation and corrected the error.

* Page 14 - The statement 'most of these were severely obese patients (59.9%, P < 0.001)' is incorrect in this context. The percentage reported here is the proportion of the severe obesity group that had insufficient PA. The largest proportion of patients with insufficient PA came from the overweight (35.7%) group, with 10.2% of the total being in the severe obesity group.

Our comments and revisions: Point taken. Thank you for your help. We agree with you that the description was not correct in the context. We have updated in the revised manuscript to highlight the proportion within each BMI group for comparison: ‘nearly half of the patients were classified as having insufficient PA (47.3%) (Table 1); severe obesity had the highest proportion (59.9%), while normal weight had the lowest (40.6%, P < 0.001).’

* Page 19 - Please change '...being obese were' to 'being obese was'

Our comments and revisions: Point taken. Thank you for your advice. We have changed the word in the text.

---

## [Decision Letter · Decision Letter 1]

15 Sep 2020

Facing obesity in pain rehabilitation clinics: profiles of physical activity in patients with chronic pain and obesity – a study from the Swedish Quality Registry for Pain Rehabilitation (SQRP)

PONE-D-19-34615R1

Dear Dr. Dong,

We’re pleased to inform you that your manuscript has been judged scientifically suitable for publication and will be formally accepted for publication once it meets all outstanding technical requirements.

Kind regards,

Adewale L. Oyeyemi, Ph.D

Academic Editor

PLOS ONE

Additional Editor Comments (optional):

Reviewers' comments:

Reviewer's Responses to Questions

**Comments to the Author**

1. If the authors have adequately addressed your comments raised in a previous round of review and you feel that this manuscript is now acceptable for publication, you may indicate that here to bypass the “Comments to the Author” section, enter your conflict of interest statement in the “Confidential to Editor” section, and submit your "Accept" recommendation.

Reviewer #1: All comments have been addressed

2. Is the manuscript technically sound, and do the data support the conclusions?

Reviewer #1: (No Response)

3. Has the statistical analysis been performed appropriately and rigorously? 

Reviewer #1: (No Response)

4. Have the authors made all data underlying the findings in their manuscript fully available?

Reviewer #1: (No Response)

5. Is the manuscript presented in an intelligible fashion and written in standard English?

Reviewer #1: (No Response)

6. Review Comments to the Author

Reviewer #1: (No Response)

7. PLOS authors have the option to publish the peer review history of their article (what does this mean?). If published, this will include your full peer review and any attached files.

Reviewer #1: No

---

## [Editor Report · Acceptance letter]

17 Sep 2020

PONE-D-19-34615R1 

Facing obesity in pain rehabilitation clinics: profiles of physical activity in patients with chronic pain and obesity – a study from the Swedish Quality Registry for Pain Rehabilitation (SQRP) 

Dear Dr. Dong:

I'm pleased to inform you that your manuscript has been deemed suitable for publication in PLOS ONE. Congratulations! Your manuscript is now with our production department. 

Kind regards, 

on behalf of

Dr. Adewale L. Oyeyemi 

Academic Editor

PLOS ONE